# Aquaporin Channels in the Heart—Physiology and Pathophysiology

**DOI:** 10.3390/ijms20082039

**Published:** 2019-04-25

**Authors:** Arie O. Verkerk, Elisabeth M. Lodder, Ronald Wilders

**Affiliations:** 1Department of Medical Biology, Amsterdam University Medical Centers, University of Amsterdam, 1105 AZ Amsterdam, The Netherlands; r.wilders@amc.uva.nl; 2Department of Experimental Cardiology, Amsterdam University Medical Centers, University of Amsterdam, 1105 AZ Amsterdam, The Netherlands; e.m.lodder@amc.uva.nl

**Keywords:** aquaporin, water channel, heart, myocyte, edema, failure, energy, electrophysiology, ion channels, Ca^2+^ transient

## Abstract

Mammalian aquaporins (AQPs) are transmembrane channels expressed in a large variety of cells and tissues throughout the body. They are known as water channels, but they also facilitate the transport of small solutes, gasses, and monovalent cations. To date, 13 different AQPs, encoded by the genes *AQP0*–*AQP12*, have been identified in mammals, which regulate various important biological functions in kidney, brain, lung, digestive system, eye, and skin. Consequently, dysfunction of AQPs is involved in a wide variety of disorders. AQPs are also present in the heart, even with a specific distribution pattern in cardiomyocytes, but whether their presence is essential for proper (electro)physiological cardiac function has not intensively been studied. This review summarizes recent findings and highlights the involvement of AQPs in normal and pathological cardiac function. We conclude that AQPs are at least implicated in proper cardiac water homeostasis and energy balance as well as heart failure and arsenic cardiotoxicity. However, this review also demonstrates that many effects of cardiac AQPs, especially on excitation-contraction coupling processes, are virtually unexplored.

## 1. Introduction

Aquaporins (AQPs) constitute a major and diverse transmembrane channel family found in most living organisms [1,2,3]. They facilitate movement of water along osmotic gradients and were originally named water channels [4]. Water constitutes approximately 70% of organism mass [5], and therefore, AQPs—together with simple water diffusion across the hydrophobic bilayer—are important for many different (patho)physiological processes (for reviews, see [6,7]).

AQPs are relatively small membrane proteins that constitute a monomer containing six transmembrane spanning domains, intracellular C- and N-termini, and a central water pore [5,8,9,10]. In the plasma membrane, four monomers form a functional tetramer, with each monomer functioning independently. Although somewhat controversial, the central tetrameric pore of some AQPs has been proposed to conduct also small solutes, monovalent anions, heavy metal ions, and gasses, i.e., O_2_ and CO_2_. AQP channels are voltage-independent and do not show gating properties. To date, 13 AQPs are known in mammals (AQP0–12, encoded by the genes *AQP0*–*AQP12*), and they are divided into three subfamilies based on their pore selectivity. Table 1 summarizes the subdivision of the mammalian AQPs and their non-controversial basic properties. In short, AQPs 0, 1, 2, 4, 5, 6, and 8 belong to the classical water selective AQPs. AQPs 3, 7, 9, and 10, also named aquaglyceroporins, are less water permeable, but they also pass small neutral solutes, such as glycerol and urea. AQPs 11 and 12, initially named AQPX1 and AQPX2, are classified to another AQP subfamily, named unorthodox aquaporins, the properties of which are less clear.

The mammalian AQPs are present in many cell types and organs, for example in kidney, brain, lung, digestive system, eye, and skin [7]. It is generally accepted that they have important (patho)physiological roles in, for example, water reabsorption in the kidney, water exchange across the blood–brain barrier, and growth and vascularity of tumors. We refer to several extensive reviews for their role in normal physiological processes as well as their pathophysiological function in the above-mentioned organs, cancer, and obesity [11,12,13,14,15,16,17,18,19]. AQPs are also found in the heart and cardiomyocytes [20,21,22], but their role in normal heart function and cardiac disorders has not intensively been studied [23]. In this review, we first provide a brief overview of the presence of AQPs in the heart. Subsequently, we highlight their (patho)physiological role, which has primarily been studied using *AQP* knock-out transgenic mice. We conclude that AQPs are not only important for cardiac water homeostasis, but may also affect cardiac excitation-contraction coupling—either directly or indirectly via dysfunction of other organs—due to their permeability to glycerol, arsenite, and other small, neutral solutes as well as to interactions with various ion channel proteins and connexins. However, despite the great progress in the determination of the characteristics and functions of AQPs in the heart, various effects of cardiac AQPs seem still unknown.

## 2. AQP Presence in the Heart

Since the first AQP, i.e., AQP1, was identified in rat hearts by Agre and co-workers [26], several other AQP subtypes have been discovered in cardiac tissue of various animal species (including mouse, rat, sheep, and goat) and human (for reviews, see [20,21,22]). The cardiac AQP presence and expression depends on various factors such as species, sex, development and aging, with data sometimes being contradictory between species [27]. For example, the presence of AQP1 is high in rat heart at embryonic days 14–16 [26], but the level of expression was substantially reduced after birth [26], and even further decreased postnatally [28]. AQP1 was found in the endocardium of the sheep fetal heart at a very early stage [29], but in contrast to rat, was higher at older developmental stages [30]. In mice, aging increased AQP1 and AQP4 levels [31]. Cardiac AQP1 expression was found to be lower in female mice [32]. Additionally, in these mice, *AQP1* knock-out resulted in increased AQPs 4, 7, and 11 expression in male, but not female, hearts. More details about the dependence of findings on AQP expression in the heart on the techniques and species of the underlying studies, can be found in the reviews of Egan et al. [20] and Rutkovskiy et al. [21]. Butler and colleagues [27] performed a large study of mRNA expression of AQPs 1–11 in hearts of different species by real-time PCR which will detect all levels of expression (even very low levels); mRNA expression of *AQPs* 1, 4, 6, 7, 8, and 11 was found in mouse hearts, whereas *AQPs* 1, 4, 6, 7, 9, and 11 were found in rat hearts, and *AQPs* 1, 3, 4, 5, 7, 9, 10, and 11 in human hearts. *AQP0* (*MIP*), *AQP12A*, and *AQP12B* were not assessed. Western blot analysis of these samples indicated cardiac expression only of AQPs 1 and 4; AQP7 was not assessed as its high expression in adipose tissue might preclude proper interpretation of the results. RNA-sequencing (RNAseq) now allows for direct quantification of relative expression levels without the inherent primer bias of RT-PCR. Large RNAseq studies of cardiac tissue are now available [33,34]. Based on these datasets, human cardiac *AQP* expression seems to consist mainly of *AQP1*, *AQP3*, and *AQP7*, the expression of the other genes being small or (nearly) absent (Figure 1).

AQP sub-cellular localization has only been studied in mouse and rat and is summarized in Figure 2. Immunohistochemistry demonstrated that AQPs 1, 4, and 7 were primarily localized to the capillary endothelium and endocardium [21,35,36]. In rat, AQP1 was found in the membrane of isolated atrial and ventricular myocytes, primary atrial myocyte cultures, and in situ myocytes of frozen atrial and ventricular tissue [37]. Thereafter, other studies confirmed the presence of AQP1 as well as AQP4 in cardiomyocytes of mouse [27,38], rat [27,35], and human [27,39]. Zhang et al. [38] demonstrated abundant expression of AQP1 and 6 at the sarcolemma of mouse cardiomyocytes. In the human heart, AQP1 has a striated pattern of staining that shows overlap with staining for the Z-lines due to its localization in T-tubules [39]. As discussed in an editorial, Miller [40] suggested that this particular localization may be important for coordination of the myocardial contraction via CO_2_ transport mechanisms rather than water transport in the T-tubules itself. In the study of Zhang et al. [38], AQP4 was exclusively found in the intercalated discs of mouse cardiomyocytes, although this is not a consistent finding [27,39]. AQP4 has a PDZ domain binding sequence at the C-terminal [41]. Therefore, AQP4 can bind to synapse-associated protein 97 (SAP97) and syntrophin [42] which, due to their localization in the intercalated disc, may be a valid explanation for the prevalent distribution of AQP4 in the intercalated discs. The consequences of such a specific sub-cellular distribution in cardiomyocytes may be critical for action potential generation and conduction, as discussed in Section 4.

As already stated above, cardiac AQP expression is affected by normal physiological conditions including sex, development, and aging. However, and even more importantly, these days various pathologies are known that may affect cardiac AQP expression and, thereby, their role in (patho)physiology of the heart. In Table 2, we provide an overview of the pathologies and conditions associated with altered AQP expression, and a detailed description of the (potential) effects are described in various sections below.

## 3. AQPs and Myocardial Edema

Interstitial water accumulation, also named edema, occurs in tissue when transcapillary water transport surpasses the flow of the lymphatic system, and consequently reduced cardiac lymphatic flow is frequently associated with myocardial edema [43,44,45]. Water accumulation in the myocardium is directly associated with contractility changes [46] and mortality due to myocardial dysfunction [47,48]. Myocardial edema may occur in pathologic conditions, including cardiac ischemic injury following myocardial infarction [49], cardiac bypass surgery [20], or severe burn [50]. In general, water can pass the lipid bilayer of the cell membrane by different processes, including diffusion, coupled to ion channels or substrate transporters, and AQPs [23]. AQPs form the major water transport mechanism in cardiomyocytes [51,52], but species differences seem to exist [20,23]. Consequent to their water channel function, AQPs are frequently involved in myocardial edema as discussed below and summarized in Table 2.

### 3.1. Myocardial Edema following Myocardial Infarction

Warth et al. [49] studied the role of AQP4 in myocyte swelling in a murine model of myocardial infarction. They found an upregulation of AQP4 mRNA expression and this correlated with the size of the infarction. Zhang et al. [38] studied the expression pattern of AQPs in the process of water accumulation and heart dysfunction following myocardial infarction in mice. They found an upregulation of AQP1, 4, and 6 in response to myocardial infarction. The AQP4 expression showed a time course that coincided with that of myocardial edema and cardiac dysfunction. However, AQP1 and AQP6 expression increased persistently up to four weeks. Li et al. [58] demonstrated that *AQP1* knock-out has cardioprotective properties in the setting of myocardial infarction. *AQP1* knock-out mice have reduced myocardial edema and a smaller cardiac infarct size as well as an improved cardiac function as determined via left ventricular catheter measurements. In addition, simvastatin reduced myocardial edema during ischemia-reperfusion by suppressing the upregulation of AQP1, 4, 8, and 9 in a partially PKA-dependent manner [59], further indicating the importance of AQPs in myocardial edema.

### 3.2. Myocardial Edema following Cardiopulmonary Bypass Surgery

Cardiac surgery may result in post-operative myocardial stunning and the development of myocardial edema (for review, see [20]). In goats, cardiopulmonary bypass surgery resulted in an increased myocardial muscle tissue water content, which was accompanied by an increased AQP1 expression [53,54]. The amount of myocardial edema was enhanced by additional AQP1 overexpression [53], while additional AQP1 knock-down and blockade reduced the severity of myocardial edema [53,54]. Cardiopulmonary bypass surgery in rats not only resulted in an increased AQP1 expression, but also in an increased AQP4 expression [55]. Interestingly, hydrogen-rich solutions, which may have therapeutic effects on various diseases [60], suppressed the increase in AQP expression and reduced myocardial injury induced by cardiopulmonary bypass surgery [55]. In addition, it has been suggested that diazoxide, an opener of mitochondrial ATP-sensitive K^+^ channels, may decrease myocardial edema during cardiopulmonary bypass surgery (see [61], and primary references cited therein). Intriguingly, diazoxide treatment in patients undergoing cardiopulmonary bypass surgery lowers the AQP7 expression in atrial tissue [61].

### 3.3. Myocardial Edema following Severe Burn

Severe burn injury may affect cardiac function, including an increase in myocardial oxygen consumption, an increase in cardiac output, and the occurrence of tachycardias [62]. Li et al. [50] studied the expression of AQP1 in rat hearts after severe burns and the effects of fluid resuscitation treatment post scalding. They found that scald injuries caused a synchronized increase in water content and *AQP1* mRNA and protein expression in cardiac tissue. They also tested the effects of fluid resuscitation, a frequently used intervention in case of severe burns [63], on myocardial edema and AQP1 expression profile. Rats receiving immediate fluid resuscitation treatment showed less severe myocardial water accumulation and lower AQP1 expression than rats receiving delayed fluid resuscitation treatment [50].

## 4. AQPs and Cardiac Electrophysiology

The pump function of the heart is due to coordinated excitation-contraction coupling in atria and ventricles [64]. Excitation is the result of cardiac action potentials, which consist of five successive phases. During each of these phases, different ionic currents are activated, which are either carried through voltage-gated Na^+^, K^+^, and Ca^2+^ channels or through specific ion exchange mechanisms (for reviews, see [65,66]). Dysfunction of cardiac ion channels and/or exchangers may, thus, affect action potential properties [65], conduction [67], and pacemaker activity [68], all potentially leading to life threatening cardiac arrhythmias and sudden cardiac death [69,70]. In contrast to the cardiac ion channels as well as pumps and exchangers, most mammalian AQP channels conduct neutral molecules and do not transport ions. However, there may be some exceptions in which AQPs conduct charged particles [5,71]. For example, AQP1 activated by cyclic guanosine monophosphate conducts monovalent cations, including Na^+^, K^+^, and Cs^+^, through the central pore of the tetramer [72,73,74,75]. Moreover, AQP6 can act as a chloride channel under specific conditions related to the presence of Hg^2+^ and low pH [76,77]. Whether such ion flows have a direct physiological relevance for cardiac electrophysiology is to our knowledge not known. Apart from acting as an ion channel, AQPs may affect cardiac electrophysiology also indirectly via swelling and/or shrinking of the myocytes and the consequent changes in ionic composition of the cytosol and the extracellular fluid. In addition, AQPs may exert an effect on cardiac electrophysiology by crosstalk with various other membrane proteins or by forming macromolecular complexes [9]. Below, we discuss these possibilities in detail. Data from *AQP* knock-out transgenic mice are summarized in Table 3.

### 4.1. AQPs and the Nernst Potential

AQPs transport water between cardiomyocytes and the extracellular space. Consequently, intracellular and extracellular ionic compositions will be affected by AQPs. It is well established [84] that a specific transmembrane ionic current (*I*) is set by the membrane conductance (*G*) of the ion of interest and its electrical driving force, which is the difference between the equilibrium potential (*E_rev_*) for the particular ion and the membrane potential (*V_m_*), as expressed by the ohmic relationship:(1)$$I=G\times \left(V_{m}-E_{rev}\right)$$

The *E_rev_* of the ion in question is determined by the ratio of the intracellular ([*X*]*_i_*) to extracellular ([*X*]*_o_*) ionic concentration [84,85], and is expressed by the Nernst equation:(2)$$E_{rev}=\left(\frac{R\times T}{z\times F}\right)\times ln\left(\frac{{\left[X\right]}_{o}}{{\left[X\right]}_{i}}\right)$$
in which *R*, *T*, *z*, and *F* denote gas constant, absolute temperature, valence, and Faraday constant, respectively. Thus, any change in intracellular and/or extracellular ionic composition due to water transport may affect ion channel activity, and conversely. Indeed, swelling and shrinking of cardiomyocytes affect various membrane currents, including the L-type Ca^2+^ current (I_Ca,L_) [86], and the rapid and slow component of the delayed rectifier K^+^ current (I_Kr_ and I_Ks_, respectively) [87].

### 4.2. AQPs and Cell Volume- and Osmotically-Activated Channels

The water transport of AQPs will also regulate cell volume and osmotic values. Importantly, the expression of AQPs itself is affected by osmolarity, thereby further drastically altering the water permeability of the cell membranes (see [88], and primary references cited therein). Cardiomyocytes have various cell volume-activated ion channels (VACs) as well as osmotically-activated cation channels. VACs are Cl^−^ or K^+^ channels activated by an increase in cell volume [89,90]. For example, swelling in ventricular and Purkinje cardiomyocytes activates an outwardly rectifying Cl^−^ current (I_Cl(swell)_), which results in resting membrane potential depolarization and action potential shortening in an osmotic gradient-dependent fashion [91,92]. Resting membrane depolarization and action potential shortening are conditions that facilitate the occurrence of reentrant arrhythmias [93]. Despite the pro-arrhythmic effects of volume changes, the direct effect of AQPs on VACs in cardiomyocytes has not yet been studied.

The Transient Receptor Potential Vanilloid 4 (TRPV4) is an osmotically-activated channel and its expression in cardiomyocytes is enhanced during ageing [94]. TRPV4 channels are permeable to Ca^2+^ ions and the influx of Ca^2+^ via these channels contributes substantially to osmotically-induced Ca^2+^-transient and contractility changes, as well as tissue damage following ischemia-reperfusion [95]. In addition, the Ca^2+^ load due to TRPV4 channel activation may lead to activation of Ca^2+^-activated K^+^ channels in the heart, as detailed in Section 4.3 below. Recently, it has become clear that AQP2, 4, and 5 and TRPV4 show interactions in various tissues (see [94], and primary references cited therein). Thus, AQPs may exert an effect on cardiac electrophysiology and arrhythmias via VAC and TRPV4 channel activation, but detailed experiments are needed to study this hypothesis.

### 4.3. AQPs and K^+^ Channels

Inward rectifier K^+^ (Kir) channels are permeable to K^+^ ions and they conduct larger inward currents (at non-physiological membrane potentials, negative to the K^+^ equilibrium potential) compared to outward currents. Various Kir subfamilies exist throughout the body [96]. AQP4 is the predominant water channel in astrocytes [97]. In such cells, AQP4 affects the ATP-sensitive inward rectifier K^+^ channel, Kir4.1, encoded by *KCNJ10*, through spatial buffering of K^+^ by facilitating the movement of water through the membrane [98,99]. In addition, genetic variation in *AQP4*, and potentially altered Kir4.1 function, are predisposing factors for sudden infant death syndrome (SIDS) [100,101]. Kir4.1 is also present in the heart, and its activation may have both anti- and pro-arrhythmic effects [102]. In addition, Kir2.x is responsible for the inward rectifier K^+^ current (I_K1_), the current responsible for setting the resting membrane potential. Kir2.x has PDZ domains similar to Kir4.1 [103], and thus Kir2.x in the heart may also be modulated by AQPs. Further research is needed to elucidate such AQP and Kir2.x interaction and potential functional effects thereof in cardiomyocytes.

As mentioned in Section 4.2 above, osmotically-induced TRPV4 activation may lead to Ca^2+^ load, which subsequently may activate various Ca^2+^-activated K^+^ channels [94]. Ca^2+^-activated K^+^ channels, especially the small-conductance Ca^2+^-activated K^+^ (SK) channel, are important for atrial repolarization and are involved in atrial fibrillation (AF), and may affect ventricular electrophysiology during heart failure [104,105]. Thus, AQPs may affect basic cardiac electrophysiology, and may promote cardiac arrhythmias through TRPV4-induced Ca^2+^-load. In addition, K_V_1.5 channels, TREK channels and TWIK channels—all present in the heart [66]—seem also closely located to AQP4 [106,107,108,109] but, so far, no studies have been performed to unravel whether these channels are molecular partners of AQP4 in cardiomyocytes.

### 4.4. AQPs and Cx43

Cell-to-cell communication through gap junction channels, which are built of connexin proteins, is essential for coordinated propagation of action potentials in the heart [110]. Relationships between connexins and AQPs are well established in vertebrate lens and brain with a decrease in connexin expression upon downregulation of AQPs (for review, see [111]). This seems also valid in cardiac tissue. Expression of connexin43 (Cx43) was downregulated in *AQP4* knock-out mice [79], but inversely correlated with AQP1 expression in goats following cardiopulmonary bypass surgery [53]. To our knowledge, the relevance of AQPs in cardiac impulse propagation or conduction has not yet been studied, neither with ECGs nor with more sophisticated techniques at tissue levels. However, a decrease in expression of Cx43, as found in the *AQP4* knock-out mice, will make the heart more vulnerable to reentry-based arrhythmias [110].

### 4.5. AQPs and Na^+^ Channels

In dorsal root ganglion neurons, *AQP1* knock-out resulted in a reduced action potential firing, reduced action potential upstroke velocities, and in a decrease of the tetrodotoxin-resistant (TTX-R) Na_V_1.8 current and expression [112]. Using immunoprecipitation studies, Zhang and Verkman [112] showed a clear AQP1–Na_V_1.8 interaction. The presence of Na_V_1.8 channels in cardiomyocytes is debated [113,114], but it has been suggested that Na_V_1.8 channels may at least affect cardiac electrophysiology via intracardiac neurons [113,115]. Indeed, blockade of Na_V_1.8 in a canine model of acute AF decreased the incidence and shortened the duration of AF [115]. The AQP1–Na_V_1.8 interaction may thus also have an impact on cardiac electrophysiology and arrhythmogenesis via intracardiac neurons. Whether AQPs have also effects on other sodium channel isoforms has only been studied to a limited extent. Neither Na_V_1.7 channel expression nor the expression of its modulatory β-subunit, β1, was affected in dorsal root ganglion neurons of *AQP1* knock-out mice [112], but data on effects of AQPs on the most abundant cardiac Na^+^ channel isoform, Na_V_1.5, are lacking.

## 5. AQPs and Cardiac Contractility

Contractions of cardiomyocytes are initiated by a massive release of intracellular Ca^2+^ from the sarcoplasmic reticulum (SR), which is known as the [Ca^2+^]_i_ transient [64,116], and multiple mechanisms contribute to the associated [Ca^2+^]_i_ cycling in cardiomyocytes (for reviews, see [116,117,118]).Ca^2+^ influx through I_Ca,L_ channels triggers release of Ca^2+^ from the SR via ryanodine-2 (RyR2) channels, resulting in the start of the [Ca^2+^]_i_ transient. This so-called Ca^2+^-induced Ca^2+^ release is importantly modulated by the organization of T-tubuli, with co-localization of I_Ca,L_ and RyR2 channels, and the open probability of RyR2 channels. The [Ca^2+^]_i_ transient amplitude depends on the Ca^2+^ content of the SR. The [Ca^2+^]_i_ transient decline is importantly due to Ca^2+^ reuptake into the SR through activity of SR Ca^2+^ ATPase (SERCA) and extrusion of Ca^2+^ into the interstitium via the sarcolemmal Na^+^/Ca^2+^ exchanger (NCX). The SERCA and NCX activity compete during the [Ca^2+^]_i_ transient. Thus, changes in SERCA function indirectly affect NCX activity and vice versa, which may result in altered SR Ca^2+^ content and [Ca^2+^]_i_ transients. The diastolic Ca^2+^ concentration is regulated by the [Ca^2+^]_i_ transient decline, and leak of RyR2 channels. Thus, changes in [Ca^2+^]_i_ cycling may affect cardiac contractility [119] and/or may be a source of cardiac arrhythmias [120].

The water transport via AQPs modulates osmotic values in the cytosol, which may affect cell contractions [121,122]. Furthermore, it will also activate TRPV4 channels (see Section 4.2), resulting in Ca^2+^ overload, which is an important source for delayed afterdepolarizations [120], a cellular mechanism underlying triggered activity arrhythmias [123]. In addition, *AQP4* knock-out (KO) mice have SERCA downregulation, resulting in increased diastolic [Ca^2+^]_i_ and increased risk for cardiac arrhythmias and failure [79]. Interestingly, isoproterenol-induced cardiac weight increase was more pronounced in *AQP4* KO compared to wild-type mice suggesting that the *AQP4* KO-induced [Ca^2+^]_i_ changes may also result in altered structural remodeling [79]. Apart from diastolic [Ca^2+^]_i_ levels, systolic [Ca^2+^]_i_ levels are also increased in cells isolated from *AQP4* KO mice [79].

Apart from changes in [Ca^2+^]_i_ cycling, structural changes may also affect cardiac contractility [82,124]. At first glance, heart morphology appeared largely normal in various AQP deficient mice [58,78,83]. However, upon more close examination, cardiac weight was increased in *AQP4* KO mice [79,80], but decreased in *AQP1* KO mice [32]. The latter was accompanied by decreased cardiomyocyte dimensions [125]. A more recent study confirmed that *AQP1* KO mice have a reduced left ventricular wall thickness and mass, but also a lower capillary density [78]. *AQP1* KO mice have a lower systolic, but not diastolic, blood pressure [32], and likely a reduced maximal cardiac output [126], thus indicating that AQPs have an important regulatory role in cardiac contractility and function.

## 6. AQPs and Cardiac Energy Balance

The heart has a very high energy demand to maintain its ionic homeostasis, contractility, and metabolic processes (for review, see [127]). It is well established that cardiomyocytes require fatty acids and glucose for energy production [127,128]. However, glycerol may also be a substrate for energy production [83,129]. AQPs of the aquaglyceroporin subfamily (Table 1) allow passage of glycerol, and may, therefore, play a role in energy regulation [129]. AQP7 is the most prominent aquaglyceroporin in the heart (Section 2), and it shows upregulation in various conditions where energy balance and/or substrate may be altered, as summarized in Table 2. AQP7 is not only upregulated in mice with streptozotocin-induced diabetes mellitus [36], but also upon 72 h of fasting [36]. Furthermore, in rat, AQP7 is upregulated in response to exercise, high-protein diets, and a combination of both [56]. The importance of AQP7 in cardiac energy balance is further supported by a study of Hibuse et al. [83]. They studied the effects of *AQP7* KO in mice under basal conditions and during pressure-overload induced by either isoproterenol infusion or transverse aortic constriction. Under basal conditions, *AQP7* KO was associated with a decrease in glycerol uptake, resulting in low cardiac glycerol and ATP content. Transverse aortic constriction and infusion of isoproterenol resulted in an accelerated and more pronounced hypertrophy, respectively, in *AQP7* KO mice. In addition, *AQP7* KO mice showed a higher mortality upon transverse aortic constriction [82]. In cardiomyocytes, AQP7 may, thus, act as a glycerol facilitator and dysfunction of AQP7 may increase the susceptibility to hypertrophy as well as mortality rate.

Recently, Rutkovskiy and co-workers linked the cardioprotective effect of *AQP4* KO during ischemia to changes in protein kinases [81]. In young (≤6 months of age), but not in old (≥1 year) mice, they found a decreased expression of AMP-dependent kinase, which is important in energy status [130]. However, mice with reduced AMP-dependent kinase seem more susceptible to ischemia- and reperfusion-induced cardiac damage [131], which makes a clear association with *AQP4* KO cardioprotection debatable. Nevertheless, the study of Rutkovskiy et al. [81] demonstrates that non-aquaglyceroporins might also modulate energy status in the heart by kinase modulation.

## 7. AQP Modifiers and the Heart

Evidence is increasing that various genetic defects and pathophysiological conditions in the body may alter AQP function. Below, we discuss various AQP mutations and variants, as well as some disease mechanisms related to AQPs that may have a direct and/or indirect effect on heart function. An overview of all phenotypes associated with AQPs is provided in Appendix A.

### 7.1. AQPs Mutations and Variants

Genetic diseases caused by mutations in aquaporins are rare [132]. Loss-of-function mutations in *AQP2* may provoke nephrogenic diabetes insipidus [133,134], a condition with excessive urine production frequently resulting in electrolyte imbalance [135]. The latter is a condition that may increase the susceptibility to arrhythmias [136]. An *AQP3* splice form called *AQP3*(∆5) results in AQP3-deficient individuals with a strongly reduced glycerol transport across the membrane of red blood cells due to the AQP3 deficiency [137]. However, the cardiac phenotype of these patients is unclear. AQP4 loss of function may result in congenital cataracts and impaired hearing [138]. SIDS is frequently associated with mutations in cardiac ion channel-related genes [139]. Common genetic variants in *AQP4* in combination with common genetic variations in *KCNJ10*, encoding the ATP-sensitive inward rectifier K^+^ channel Kir4.1, have been associated with SIDS [100,101], but the genetic evidence of the studies by Opdal et al. [100,101] is limited. Recently, another AQP4 aquaporinopathy was proposed by Berland et al. [140]. In a male patient, they observed a de novo missense variation Ser111Thr in *AQP4*. Apart from intellectual disability, hearing loss, and progressive gait dysfunction, cardiac hypertrophy was detected at the age of three months, which persisted through childhood, but was completely resolved at age 16. The de novo missense variation did not affect the water permeability or protein stability, indicating that the observed disorders are not due to impaired water transport through the AQP4 channel.

Missense mutations in *AQP5* were identified in patients with inherited palmoplantar keratodermas [141], but this AQP subtype is not present in the heart (see Section 2). A common polymorphism in the promoter region of the human *AQP7* gene, resulting in AQP7 downregulation, is pathogenic for obesity and/or type 2 diabetes [142], while missense mutations in *AQP7* are associated with an inability to increase the plasma glycerol during exercise [143]. Direct effects of variants in *AQP7* on the heart are unknown, but obesity and diabetes themselves may have various adverse effects on cardiac (electrophysiological) function [144,145]. In Appendix A we provide an overview of all data available in human and in animal models. The Database of Genotypes and Phenotypes (dbGaP) [146] shows that common variants located in the vicinity of *AQP3*, *AQP7*, and *AQP9* have been associated with blood pressure and hypertension. However, these associations do not reach the threshold for genome wide significance and have not (yet) been replicated. Furthermore, mice with homozygous loss-of-function mutations in *AQP11* suffer from unexpected sudden death between two and three weeks of age [147], attributed to renal failure, but the cause for the sudden nature of the death of these animals is not investigated and could be arrhythmogenic in nature.

### 7.2. AQPs and Pathophysiological Conditions

Arsenic is a naturally occurring metalloid that unfortunately can result in global health problems affecting many millions of people [148,149]. Among other symptoms, arsenic toxicity is characterized by cardiovascular diseases [148], including peripheral arterial disease, stroke, and coronary heart disease (see [149], and primary references cited therein). The heart comprises AQPs that are permeable to arsenite (see Section 2 and Table 1). The passage of arsenite through AQPs may, thus, be responsible for intracellular arsenic accumulation and toxicity. Indeed, AQP9 knock-down in hepatocytes results in a decrease of inorganic arsenic uptake, whereas overexpression of AQP9 in hepatocytes results in an increase of inorganic arsenic uptake, leading to the enhancement of arsenic-induced cytotoxicity [148]. The effects of arsenic in relation to AQPs in the heart have only been studied to a limited extent. AQPs 3, 7, and 9 can pass arsenite (Table 1). AQP9 is absent in the human heart, whereas the expression of AQP3 is relatively low (Figure 1). No studies on AQP7 and arsenic cardiotoxicity are known so far. AQP9 is also absent in the mouse heart [27]. Interestingly, however, AQP9-null mice have 10–20 times higher arsenic concentrations in the heart than wild-type mice after injection of sodium arsenite (NaAsO_2_) [150]. In addition, they display severe bradycardia within hours after injection. AQP9 is abundantly present in the liver, and not in heart, suggesting that AQP9 affects cardiac function due to reduced arsenic clearance by the liver [150].

Lithium, which is widely and effectively used in the treatment of bipolar disorders [151], causes marked downregulation of AQP2 expression [152]. It frequently results in nephrogenic diabetes insipidus, similar to loss-of-function mutations in *AQP2* (see Section 7.1). Lithium has various side effects on cardiac function. It may, for example, induce Brugada syndrome [153] and bradycardia [154], but whether these are due to AQP dysfunction per se or due to (in)direct effects on ion channel function is unknown.

Cardiac hypertrophy induced by pressure-overload lowers AQP1 expression in rat cardiomyocytes [35]. Furthermore, cardiac hypertrophy and heart weight increase are more pronounced in *AQP7* knock-out mice upon isoproterenol infusion or transverse aortic constriction compared to wild-type mice (see also Section 6). In addition, chronic heart failure (CHF) is characterized by an upregulation of AQP2 in kidneys [155], resulting in abnormal renal water retention [156]. Administration of a vasopressin V2 receptor antagonist prevents CHF-induced AQP2 upregulation and consequently improves diuresis [155]. AQP2 modulation in the kidneys is, thus, a useful, but indirect, target for improvement of clinical status and, possibly, outcome of CHF patients [157].

Infective endocarditis, an infection of the endocardial surface of the heart, is a severe and a potentially lethal disease [158]. AQP9 is expressed in valvular tissue of patients with infective endocarditis [57], and this expression is associated with the development of acute heart failure [159]. The exact diagnosis of infective endocarditis is often missed or only made late in the course of the disease [160], and therefore, it is suggested that AQP9 expression may be a potential prognostic marker in infective endocarditis [159].

Dehydration is a severe problem in the pediatric [28] and elderly [161] population, but also in critically ill patients [162], being one of the main causes of mortality. Netti and colleagues [28] demonstrated that rats aged 50 days showed upregulation of cardiac AQP1 expression and appearance of AQP1 in cardiomyocyte plasma membranes in response to water restriction. However, such effects were absent in water-restricted rats of 25 days old. Interestingly, water restriction induced a decrease in water content at day 25, but not at day 50, indicating that the AQP1 upregulation and presence in cardiomyocyte plasma membranes prevented severe changes in cardiac water homeostasis.

Duchenne muscular dystrophy (DMD) is an X-linked disorder due to mutations in the *DMD* dystrophin gene and presents in early childhood with proximal muscle weakness [163]. DMD unavoidably leads to congestive heart failure and arrhythmias [164]. Interestingly, DMD is associated with a reduction in AQP4 expression in myofibers [165,166]. Whether this also occurs in cardiomyocytes is so far unknown, but it is tempting to speculate that AQP4 dysfunction contributes to DMD-induced cardiomyopathy and arrhythmias (see also Section 4).

## 8. Concluding Remarks

The multiple essential roles that AQPs play in cardiac water homeostasis are evident from the changes in myocardial edema in pathophysiological conditions. The effects of AQPs on cardiac electrophysiology are largely unexplored. However, as discussed above, AQPs have tight protein–protein interactions with various membrane ion channels and connexins as well as effects on [Ca^2+^]_i_ cycling, all important for proper cardiac excitation. It is, therefore, highly susceptible that AQPs are important determinants of both proper cardiac electrophysiology and arrhythmogenesis, but detailed studies are required to test this hypothesis. AQP dysfunction may reduce blood pressure and maximal cardiac output, which may be due to reduced cardiac contractility consequent to changes in [Ca^2+^]_i_ cycling and cardiomyocyte and heart morphology. Further research is required to elucidate the exact mechanism of the reduced pump function of the heart. AQPs, especially AQP7, which is a member of the aquaglyceroporin subfamily and is abundantly expressed in the human heart, seem to have a role in cardiac energy balance. AQP7 acts as a glycerol facilitator and glycerol is a substrate for energy production. AQP mutations and variants may have a direct or indirect effect on cardiac function, while some pathophysiological conditions, e.g., arsenic toxicity and CHF, may be due to the dysfunction of AQPs. Furthermore, the use of specific medications, like lithium in case of bipolar disorders, may result in dysfunction of AQPs and improper cardiac functioning. Despite the impact of AQPs on proper cardiac water homeostasis, energy balance, heart failure, and arsenic cardiotoxicity, and their potential effects on cardiac excitation-contraction coupling, the cardiac phenotype of loss of function of specific AQPs is not always immediately clear. In this respect, it is worth noting that this is likely due to the presence of other water-specific AQPs, which may take over cardiac water transport function [27]. In fact, the total expression of AQPs may even be increased in case of dysfunctioning of one of the AQP types. Thus, compensatory (functional) upregulation of AQP expression and associated takeover of AQP functioning may prevent a clear phenotype in the heart. In conclusion, this review implies a critical role of AQPs in normal and pathophysiological cardiac function, but also shows that much work remains to be done to elucidate their exact role in electrical and mechanical cardiac functioning.

## Figures and Tables

**Figure 1 ijms-20-02039-f001:**
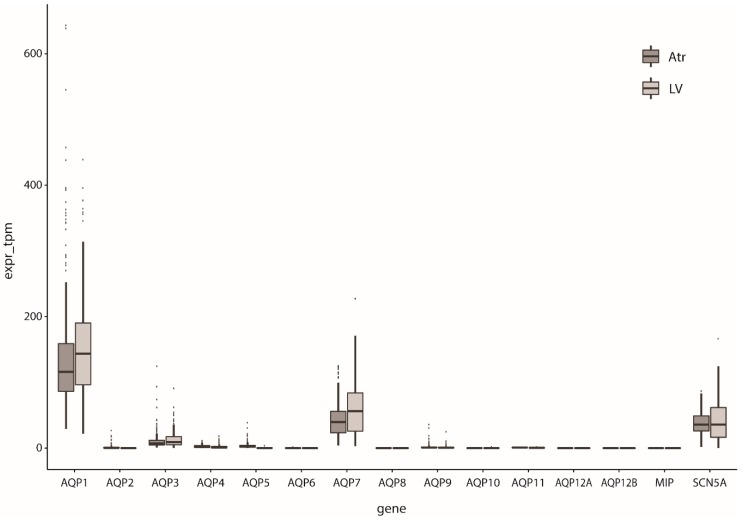
Relative expression of AQPs in human left ventricular (LV) and atrial tissue (Atr). The main cardiac sodium channel SCN5A is shown for comparison; expr_tpm indicates the number of transcripts detected for each gene per million transcripts, normalized for gene length. Expression data obtained from [33,34].

**Figure 2 ijms-20-02039-f002:**
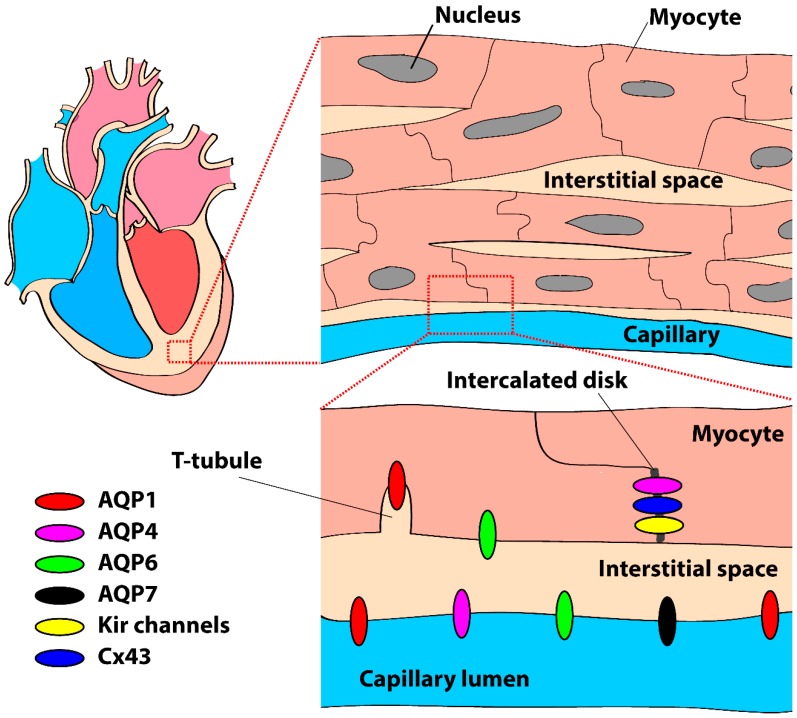
Schematic drawing of the presence and localization of aquaporins (AQPs) in the mammalian heart and cardiomyocytes, primarily based on data from mouse and rat (see body text). Kir channels = inward rectifier K^+^ channels, Cx43 = connexin43.

**Table 1 ijms-20-02039-t001:** Summary of mammalian AQPs and their basic properties.

HGNC Gene Symbol	Synonym	Subfamily	Permeability
*AQP1*	*CHIP28*	water-specific channels	H_2_O, CO_2_
*AQP2*	*WCH-CD*	water-specific channels	H_2_O
*AQP3*	*GLIP*	aquaglyceroporins	H_2_O, urea, glycerol, NH_3_, arsenite
*AQP4*	*MIWC*	water-specific channels	H_2_O
*AQP5*	–	water-specific channels	H_2_O
*AQP6*	*AQP2L*	water-specific channels	H_2_O, NH_3_, anions
*AQP7*	*AQPap*	aquaglyceroporins	H_2_O, urea, glycerol, NH_3_, arsenite
*AQP8*	–	water-specific channels	H_2_O, urea, NH_3_
*AQP9*	–	aquaglyceroporins	H_2_O, urea, glycerol, NH_3_, arsenite
*AQP10*	–	aquaglyceroporins	H_2_O, urea, glycerol
*AQP11*	*AQPX1*	unorthodox aquaporins	H_2_O
*AQP12A*	*AQP12; AQPX2*	unorthodox aquaporins	H_2_O
*AQP12B*	*INSSA3*	unorthodox aquaporins	H_2_O
*MIP*	*AQP0*	water-specific channels	H_2_O

CHIP28 = channel-forming integral membrane protein of 28 kDa; WCH-CD = water channel- collecting duct; GLIP = glycerol intrinsic protein; MIWC = mercurial-insensitive water channel; AQP2L = AQP2-like; AQPap = aquaporin adipose; AQPX1 and AQPX2 = aquaporin-like channels of a new subfamily; INSSA3 = insulin synthesis associated 3; MIP = major intrinsic protein of lens fiber. Modified from [9,10,24,25].

**Table 2 ijms-20-02039-t002:** Effect of various abnormalities on AQP expression in the heart.

Abnormality	Effect	Species	Study
Myocardial edema following myocardial infarction	AQP4 ↑	Mouse	Warth et al. [49]
	AQP1 ↑ AQP4 ↑ AQP6 ↑	Mouse	Zhang et al. [38]
Myocardial edema following cardiopulmonary bypass surgery	AQP1 ↑	Goat	Yan et al. [53], Ding et al. [54]
	AQP1 ↑ AQP4 ↑	Rat	Song et al. [55]
Myocardial edema following severe burn	AQP1 ↑	Rat	Li et al. [50]
Diabetes mellitus	AQP7 ↑	Mouse	Skowronski et al. [36]
Fasting	AQP7 ↑	Mouse	Skowronski et al. [36]
Water restriction	AQP9 ↑	Rat	Netti et al. [28]
Exercise and/or high-protein diets	AQP7 ↑	Rat	Palabiyik et al. [56]
Cardiac hypertrophy by pressure overload	AQP1 ↓	Rat	Zheng et al. [35]
Infective endocarditis	AQP9 ↑	Human	Benoit et al. [57]

**Table 3 ijms-20-02039-t003:** Observations in the heart from studies on *AQP* knock-out mice.

Knocked-Out Gene	Observed Effect	Study
*AQP1*	Increased AQP4, 7, and 11 expression in male, but not female hearts	Montiel et al. [32]
*AQP1*	Decrease in cardiac weight and cardiomyocyte dimensions	Montiel et al. [32]
*AQP1*	Lower systolic, but not diastolic, blood pressure	Montiel et al. [32]
*AQP1*	Reduced left ventricular wall thickness and mass, and a lower capillary density	Yang et al. [78]
*AQP1*	Reduced myocardial edema and a smaller cardiac infarct size upon myocardial infarction *)	Li et al. [58]
*AQP4*	Downregulated expression of Cx43 and SERCA	Cheng et al. [79]
*AQP4*	Increased diastolic and systolic [Ca^2+^]_i_ levels	Cheng et al. [79]
*AQP4*	Increase in cardiac weight	Cheng et al. [79], Cheng et al. [80]
*AQP4*	Changes in protein kinases	Rutkovskiy et al. [81]
*AQP7*	Higher mortality upon transverse aortic constriction	Gerdes et al. [82]
*AQP7*	Decrease in glycerol uptake and low cardiac glycerol and ATP content	Hibuse et al. [83]

* Similarly, it had been observed that additional AQP1 overexpression enhanced the amount of myocardial edema [53], whereas additional AQP1 knock-down and blockade reduced the severity of myocardial edema [53,54]. Cx43 = connexin43; SERCA = sarcoplasmic reticulum Ca^2+^ ATPase.

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
