# Peer review of "Aquaporin Channels in the Heart—Physiology and Pathophysiology"

_ijms, 2019, doi:10.3390/ijms20082039_

Reviewer 1 Report

The present manuscript reviewed the possible involvement of AQPs in normal and pathological heart function. After a complete description of the AQP expression in the heart, the authors have examined the AQP modifications in myocardial edema. Then the possible involvement of AQPs in electrophysiology, in cardiac contractility and in energy balance was considered. A large amount of literature was examined and described in a clear way. The figures help the reader even if not strictly expert in this field. The review is well written, attractively presented and updated.

Paragraph 4 should be shortened: cardiac action potential and contractility are well known and present in physiology textbooks.

Author Response

We thank the reviewer for the time and effort spent on reviewing our manuscript and for the positive comments and constructive advice.

We agree with the reviewer’s comment on Section 4 and have now adapted this section accordingly.

Reviewer 2 Report

The manuscript reviewed the roles of AQPs in the heart with recent new findings. The field is still growing and may need new hypothesis to stimulate the research. However, the review is full of many one-sided speculations which should be reorganized before publication.

1. The title is not clear. In fact, the authors advocate the importance of AQPs in the heart. “Aquaporins in the heart: physiology and pathophysiology” can be an example.

2. The review can be more concise by deleting the most of the introduction and the introduction of chapter 4 including Fig.3 which is mostly irrelevant to the discussion of AQPs.

3. The authors seemed to be not bothered by the normal phenotypes of AQP null mice and human genetic diseases. Taking up many one-sided mice studies by limited investigators may be unfair for the review article as they include the publication bias.

4. The issue of muscle volume contraction and the intracellular ion concentration change is not unique to the heart and the role of AQP4 in the skeletal muscle is small as water movement may not affect the local ion concentrations in normal condition.

5. As heart has lymphatic system, the roles of AQPs may affect the function of lymphatics. In fact, the role of lymphatic vessels has been reported in coronary vasospasm (PMID:30816801). By the way, the role of AQPs in the coronary circulation had little attention in this review although most AQPs are localized there. 

Author Response

We thank the reviewer for the time and effort spent on reviewing our manuscript and for the positive comments and constructive advice. As pointed out in detail in our response to comment #3 of the reviewer, we respectfully disagree with the term ‘many one-sided speculations’, which seems due to the many available AQP knock-out transgenic mice studies rather than bias by the authors.

1. The title is not clear. In fact, the authors advocate the importance of AQPs in the heart. “Aquaporins in the heart: physiology and pathophysiology” can be an example.

We thank the reviewer for addressing this issue. Our initial title was meant as a stimulus for researchers, especially in the field of cardiac excitation and contraction, to focus more on the (dys)function of cardiac aquaporins. Because the original title could also be explained with some negative connotation, we agree with the reviewer to adapt the title according to his/her suggestion.

2. The review can be more concise by deleting the most of the introduction and the introduction of chapter 4 including Fig.3 which is mostly irrelevant to the discussion of AQPs.

We partially agree with the reviewer. In the revised manuscript, we have shortened the introduction of Section 4 and removed Figure 3 completely, according to the reviewer’s suggestion. However, the main introduction of the manuscript (Section 1) is largely unchanged, because we think that some basic introduction about aquaporins, and especially the importance in many (patho)physiological processes, is needed for researchers and clinicians interested in cardiac (dys)function. 

3. The authors seemed to be not bothered by the normal phenotypes of AQP null mice and human genetic diseases. Taking up many one-sided mice studies by limited investigators may be unfair for the review article as they include the publication bias.

We respectfully disagree with the reviewer. We have written an unbiased review using the available papers about specific cardiac function of particular aquaporin isoforms. We recognized already that such studies were frequently performed using AQP knock-out transgenic mice, as was clearly mentioned in the ‘Introduction’ section (Section 1) at lines 53–54 of the original manuscript (lines 54–55 of the revised manuscript). In addition, in the ‘Concluding remarks’ section (Section 8; lines 454-461) of the old manuscript (Section 8; lines 447–454 of the revised manuscript), we addressed the difficulties with the normal function of aquaporins by the sentences: “Despite the impact of AQPs on proper cardiac water homeostasis, energy balance, heart failure, and arsenic cardiotoxicity, and their potential effects on cardiac excitation-contraction coupling, the cardiac phenotype of loss of function of specific AQPs is not always immediately clear. In this respect, it is worth noting that this is likely due to the presence of other water-specific AQPs, which may take over cardiac water transport function [26]. In fact, the total expression of AQPs may even be increased in case of dysfunctioning of one of the AQP types. Thus compensatory (functional) upregulation of AQP expression and associated takeover of AQP functioning may prevent a clear phenotype in the heart.”

4. The issue of muscle volume contraction and the intracellular ion concentration change is not unique to the heart and the role of AQP4 in the skeletal muscle is small as water movement may not affect the local ion concentrations in normal condition.

We agree with the reviewer. Under normal cardiac conditions and with unaffected aquaporin expression, water movement in cardiac tissue will be in equilibrium, leaving local ion concentrations unaltered. However, ion concentrations may be affected under various pathophysiological conditions (as now summarized in the new Tables 2 and 3 and explained in detail in various sub-sections), which results in dysfunction of aquaporins and consequently altered water homeostasis. 

5. As heart has lymphatic system, the roles of AQPs may affect the function of lymphatics. In fact, the role of lymphatic vessels has been reported in coronary vasospasm (PMID:30816801). By the way, the role of AQPs in the coronary circulation had little attention in this review although most AQPs are localized there. 

The reviewer is right. We now expanded Section 3 (pages 5 and 6 of the revised manuscript) to stress the importance of cardiac lymphatics for myocardial edema. We did not include the role of lymphatic vessels in coronary vasospasm, especially because the role of AQPs in that process is not known. In addition, reviewing the role of AQPs in the circulatory system was not the scope of our review. However, we now included in Section 3 the study of Li et al. (DOI: 10.1016/j. ijcard.2012.06.121), which further stresses the importance of AQPs in myocardial edema.

Reviewer 3 Report

Dear authors

The argument regarding the possible role of AQPs on different aspects of cardiac function in health and in disease is of real interest. In the complex, the manuscript could give an interesting contribute about the particular AQPs involvement at cardiovascular level considering the literature in continuing growth regarding this topic.  

However, the manuscript require an important revision to be reorganized in a more organic way. The authors should evidence with primary instance the expression and the role of AQPs in physiological conditions possible evidencing the main factors that can modulate, modify or interfere with their expression.

Secondarily the authors should authors should explain all of conditions in which at cardiovascular level AQPs expression is changed and the possible mechanisms of regulation. On both these aspects, a scheme or a figure could be added to help the reader in the understanding of AQPs role at the heart level.

In addition, the authors should insert at various levels of dissertation some omitted references (1. Tie et al., Aquaporins in Cardiovascular System. p. 105-113. Adv. Exp. Med. Biol. 2017; 2. Netti et al., Effects of nitric oxide system and osmotic stress on Aquaporin-1 in the postnatal heart, Biomedicine & Pharmacotherapy 2016, 81, 225–234; 3.  Zhao and Sun, Aquaporin in the proliferation and apoptosis of diabetic myocardial cells, Genetics and Molecular Research 2015, 14, 17366-17372).         

Author Response

We thank the reviewer for the time and effort spent on reviewing our manuscript and for the positive comments and constructive advice. 

However, the manuscript require an important revision to be reorganized in a more organic way. The authors should evidence with primary instance the expression and the role of AQPs in physiological conditions possible evidencing the main factors that can modulate, modify or interfere with their expression.

Secondarily the authors should authors should explain all of conditions in which at cardiovascular level AQPs expression is changed and the possible mechanisms of regulation. On both these aspects, a scheme or a figure could be added to help the reader in the understanding of AQPs role at the heart level.

We agree with the reviewer about his/her two points raised. We now included a new table (Table 2) in which we summarize the aquaporin expression changes in pathophysiological conditions. This table, and the associated new text, will also increase the clarity of the presented various sub-sections, thereby resulting in a more well-structured manuscript.

In addition, the authors should insert at various levels of dissertation some omitted references (1. Tie et al., Aquaporins in Cardiovascular System. p. 105-113. Adv. Exp. Med. Biol. 2017; 2. Netti et al., Effects of nitric oxide system and osmotic stress on Aquaporin-1 in the postnatal heart, Biomedicine & Pharmacotherapy 2016, 81, 225–234; 3.  Zhao and Sun, Aquaporin in the proliferation and apoptosis of diabetic myocardial cells, Genetics and Molecular Research 2015, 14, 17366-17372).

We have included the first two mentioned references according to the reviewer’s suggestion. However, we did not include the third reference suggested by the reviewer, mainly because it was not clear which ‘isoform’ was studied and how the diabetic myocardial cells were treated/transfected with aquaporin. Furthermore, we added a new paragraph describing the effects of water restriction on AQP1 in detail (page 12, lines 417–424).

Round  2

Reviewer 3 Report

Dear authors the manuscript was improved by the suggestions by the reviewers. Now many aspects regarding Aqp expression in the heart in physiological and patho-physiological conditions are more clear for the readers.